
# Evaluation of the Search and Rescue Leeway model into the Tyrrhenian sea: a new point of view

Antonia Di Maio[1], Mathew Vayalumkal Martin[2], and Roberto Sorgente[3]

[1]Marine Technology Research Institute INSEAN, Italian National Research Council, Rome, Italy
[2]Centre for Oceans, Rivers, Atmosphere and Land Sciences, Indian Institute of Technology Kharagpur, West Bengal, India
[3]Institute for Coastal Marine Environment IAMC, Italian National Research Council, Oristano, Italy

*Correspondence to:* Antonia Di Maio (antonia.dimaio@cnr.it)

**Abstract.** The trajectories prediction of the floating objects above the sea surface represents an important task in the search and rescue (SAR) operations. In this paper we show how may be possible estimate the most probable search area by means of a stocastic model, schematizing appropriately the shape of the object and evaluating the forces acting on it. The LEEWAY model, a Montecarlo-based ensemble trajectory model, has been used; here not only the statistical law to calculate the *leeway*

is employed but also an *almost* deterministic law inspired by the boundary layer theory. The model is nested with the subregional hydrodynamic model TSCRM (Thyrrenian Sicily Channel Regional Model) developed in the framework of PON-TESSA (National Operative Programs-TEchnology for the Situational Sea Awareness) project. The principal objective of the work is to validate the new approach of *leeway* calculation relying on a real event of Person in Water (PIW), occurred on July 2013 in the Thyrrenian Sea. The results show that assimilating a human body to a cylinder and estimating either the transition

from laminar to turbulent boundary layer and the drag coefficients, may be possible to solve a forces balance equation which permits to estimate with good approximation the search area. This new point of view leads to the possibility to check the same approach also on other different categories of targets, so as to overcome in the future the limitations associated with calculation of *leeway* by means of the standard statistical law.

**Keywords.** LEEWAY Model, *leeway*, Person In Water, Search and Rescue

## 1  Introduction

Meteocean and environmental forecast are being increasingly used in operational decision making in the sea for demographic, geographic and strategic applications. Safety and assets at sea is a shared objective of all the countries and it induces the need to have efficient ocean forecast system in order to improve prediction of sea and provide useful environmental ocean informations for increasing the effectiveness of the search operation (Breivik et al., 2013). In the event of accident, timely Search and

Rescue (SAR) intervention is helpful in significantly lowering the loss of life and also to contain the dammage. Currently a considerable amount of resources is invested in maintaining SAR capabilities by major maritime nations. However on many occasions, the available SAR capabilities prove to be inadequate to provide timely assistance in distress scenarios. Nevertheless, the effectiveness of the availabe SAR capabilities can be increased if we exploit the high-quality environmental forecast data



for SAR planning. An example of operational ocean forecasting and services coupled with Search and Rescue (SAR) activities is represented by GODAE BLUElink operational ocean prediction system (Brassington et al., 2007), which is used by the Australian Maritime Safety Authority. A list of global and regional operational ocean forecasting systems, supported by the Global Ocean Data Assimilation Experiments - GODAE (Bell et al., 2009), can be found in Davidson et al. (2009), while a

recently review of the evolution of the global and regional forecasting system from GODAE into GODAE OceanView (Bell et al., 2015) is described in Tonani et al. (2015). These systems are based on ocean general circulation models (OGCM) and data assimilation techniques that are able to correct the model with informations inferred from different types of observations. The first step in marine search planning is to determine the most probable area containing the searched object and that is even more important if the target is person in water (PIW). The definition of the probable search area is essentially linked

to the quantification of some unknowns, such as the Last Known Position (LKP) and typology, shape and dimensions of the objects. Moreover, a PIW or object without propulsion is subject also to drift from ocean currents, wave action and direct wind action (Davidson et al., 2009). Hence, an effective SAR planning requires accurate environmental forecasting, especially regarding surface currents, temperature and surface winds for short range time-scale. Today, these informations can be provided in near real time (NRT) from global and regional operational oceanography systems following on from of the successful

implementation of the Global Ocean Observing System (GOOS).

In the Mediterranean Sea, operational forecast starting from the year 2000 is provided by the Mediterranean Forecasting System - MFS (Tonani et al., 2008). The MFS uses a horizontal grid of resolution of 1/16° and provides detailed forecast at a regional scale for the whole Mediterranean basin. The fundamental part of the system is the assimilation scheme for the blending of the observations into the model in order to provide the most accurate description of the past and the best initial condition for the

forecast fields at large scale (Oddo et al., 2009).

In many cases, SAR activities may also require fields of interest on a high spatial resolution. Sub-regional forecast systems that can resolve small-scale processes and fronts characteristics of the study area are necessary. A numerical technique widely used in operational oceanography to increase the horizontal resolution is the downscaling procedure, which permits to adapt the large scale features to the coastal areas by means of model domains cascade embedded (Pinardi et al., 2003). This has been

achieved through implementation of a nested high resolution ocean model for the central Mediterranean basin developed in the framework of the project Development of TEchnology for Situational Sea Awareness (TESSA). This hydrodynamic model, named Tyrrhenian and Sicily Channel sub-Regional Model (TSCRM), is based on POM (Blumberg and Mellor, 1987) and has a horizontal resolution of 1/48°, then three time MFS.

TESSA project is supported by PON "Ricerca e Competivita' 2007-2013" program of the Ministry for Education, University

and Research. The general objective is to improve and consolidate the products and the operational oceanography service in Southern Italy and to integrate this service with technological platforms of dissemination of information for the Situational Sea Awareness (SSA) in support also to the SAR activities on the sea. In Italy these activities are ensured from the Coast Guard, under the competence of the minister of the infrastructures and transport of Italian government.

In this paper we present some numerical results about the prediction of PIW trajectories in the central Tyrrhenian Sea. We use

two different approaches for the *leeway* calculation: the first approach is standard and it consists in the *leeway* calculation by



means of the statistical parameters tabled by Allen and Plourde (1999); the second one is a variant respect to the precedent and it consists in the *leeway* calculation by means of the forces balance equation based on modifications of the target boundary layer. The objective of this work is to validate latter approach but we hope that in future the new method permit to define a practical formula of *leeway* calculation helpful not only in similar PIW cases but also in other targest cathegories.

5    The new approach is embedded in the LEEWAY model, a drift model based on a stochastic approach (Breivik and Allen, 2008), combined with environmental forecast data from TSCRM and European Centre for Medium Whether Forecast (ECMWF). The ensuing sections of the paper are organized as follows: section 2 describes the models used and the numerical experiments; an analysis of the model results is presented in section 3 together with possible future extensions. A summary and concluding remarks are given in section 4.

## 2    Material and Methods

In this section we describe our drift forecast operational numerical model, developed to predict the trajectories of floating objects (SubSection 2.1), the characteristics of the LEEWAY model including also our variation about the calculation fo the *leeway* (SubSection 2.2) and the different experiments executed (SubSection 2.3) based on Person In Water (PIW).

### 2.1    Ocean Model

The current forecasting model used in this work is the Tyrrhenian and SiCily strait sub-Regional Model (TSCRM), an operational sub-regional, nested ocean model implemented into the central Mediterranean Sea during the framework of the project TESSA. TSCRM is a free surface three-dimensional primitive equation finite difference hydrodynamic model, based on the Princeton Ocean Model - POM (Blumberg and Mellor, 1987). It solves the equations of continuity, motion, conservation of temperature, salinity and assumes that the fluid is hydrostatic and the Boussinesq approximation is valid. The density is cal-

culated by and adaptation of the UNESCO equation of state revised by Mellor (1991); the horizontal viscosity and vertical mixing coefficients are computed respectively using the Smagorinsky approach (Smagorinsky, 1993) and the Mellor and Ya-mada (1982) turbulence closure scheme. It is an extension of the Sicily Strait sub-Regional Model implemented by Sorgente et al. (2011), and successively made operational by Fazioli et al. (2015) and embedded into the regional model of the MFS (Tonani et al., 2008) through the downscaling procedure (Sorgente et al., 2003). This modelling technique, widely used to

produce numerical weather prediction (Koch and McQueen, 1987), permits to downscale the model results from the regional scale to the sub-regional scale by transmitting values of temperature, salinity and velocity (three-dimensional fields daily mean) at the interconnecting boundaries from the coarse resolution grid to the fine resolution. That permits a more detailed description of the circulation in those areas where the regional model cannot resolve the small mesoscale features. By means of the downscaling technique, TSCRM receives informations at lateral open boundaries by an off line, one-way asynchronous nesting

technique (Zavatarelli et al., 2002), coupling to the regional model which it is embedded. Surface momentum and buoyancy fluxes are interactively computed by means of standard bulk formulae using predicted model Sea Surface Temperature and 6-h (00:00, 06:00, 12:00, 18:00 UTC) and 0.25° atmospheric variables provided by the European Centre for Medium Range





Weather Forecast operational analyses (Sorgente et al., 2011). TSCRM produces 1-h and 1/48° fields that are used toghether with the 6-h and 0.25° wind velocity fields to force the lagrangian model LEEWAY.

## 2.2 LEEWAY model

The drift of a floating object above the sea surface is the result of the balance between the hydrodynamic and aerodynamic forces. This balance induces the object to move with a certain angle respect to downwind direction (Richardson, 1993; Breivik and Allen, 2008), then it is foundamental to take into account the *leeway* to estimate with good approximation the trajectories of drifting objects.

The term *leeway* refers to object's motion induced by the atmospheric wind (10 m reference height) and waves relative to ambient current (between 0.3 and 1.0 m depth). This definition standardized the reference levels for the measurements of *leeway* for SAR objects and provides a practical way to utilize current and wind vectors from numerical models of measurements (Allen, 2005). The empirical relation between *leeway* and wind speed is given by empirical coefficients (linear regression coefficients and their standard deviations) for a total of 63 different targets classes, tabled as a result of field campaigns performed by the U.S. Coast Guard (Allen and Plourde, 1999). These coefficients give an estimate of the relation between the wind speed and *leeway* velocity vector, converted into following components: a downwind component and positive (right of the wind) and negative (left of the wind) crosswind *leeway* components (Allen, 2005); they are explicited in the following equations:

$$\boldsymbol{L_d} = (a_d + \epsilon_d)\boldsymbol{W_{10m}} + b_d + \epsilon_d \tag{1}$$

$$\boldsymbol{L_c^+} = (a_c^+ + \epsilon_c)\boldsymbol{W_{10m}} + b_c^+ + \epsilon_c \tag{2a}$$

$$\boldsymbol{L_c^-} = (a_c^- + \epsilon_c)\boldsymbol{W_{10m}} + b_c^- + \epsilon_c \tag{2b}$$

The wind velocity vector $\mathbf{W}_{10m}$ is the wind speed measured at $10m$ reference height, or simulated by a weather forecast model. The term $L_d$ of equation (1) represents the downwind *leeway* component (DWL), while the terms $L_c$ of equation (2a) and (2b) are the crossway *leeway* components (CWL), that are the divergence of the SAR object from the downwind direction. The parameter $a$ is denoted the *leeway* rate (*leeway* to wind ratio), $b$ is the regression intercept at zero wind and $\epsilon$ the regression residual (Breivik et al., 2012). The determination of the *leeway*, then, more precisely the relation between the wind speed and the *leeway* speed and divergence angle, gives a measure of how much wind directly pushes on a object floating at the sea surface.

It is necessary to assume that the empirical coefficients of linear regression of the equations 1 and (2a) and (2b) include also the contribute of the Stokes drift (Breivik and Allen, 2008). Then in our experiments we assumed that *leeway* can be expressed as a function of the wind only and we use a practical definition of *leeway* which doesn't distinguish between the wind and the wave influence; this approximation can be extend also to other small objects and to most sea states, according to Breivik et al.


(2012).

Estimating the DWL and CWL *leeway* linear regression coefficients, it is possible to recreate the expected drift of an object by means of modelled or measured data concerning the wind and sea current. Because the LEEWAY model uses empirical formulae and imperfect approximation to the hydrodynamic laws, a Monte Carlo technique works to generate an ensemble

through random perturbations; the ensemble permits to take into account the uncertainties about the forcings (wind and current), *leeway* drift properties (draught, length and beam), and the Last Know Position (LKP) of the search object. The trajectory is then obtained by estimating at each time step the corresponding probability density function of containment of the object, and its envelope is the estimation of the search area (Davidson et al., 2009).

An exhaustive description of the stochastic approach used into the LEEWAY suite is given by Hackett et al. (2006).

In this work we present a variation about the calculation of the *leeway* replacing the linear regression equation described by equations (1), (2a) and (2b) with an *almost* (due to uncertainties in forcing fields) deterministic law which is the balance of the aerodynamic and hydrodynamic forces acting on the target, evaluating the modification of its boundary layer; the target is supposed approximately symmetrical, then the lift can be negleted and we suppose that it is tracked by the surface current under the hypothesis of steady motion:

$$\boldsymbol{F_P} + \boldsymbol{F_A} + \boldsymbol{F_{ADw}} + \boldsymbol{F_{ADa}} = \boldsymbol{0} \tag{3}$$

where $\boldsymbol{F_P}$ is the weight force, $\boldsymbol{F_A}$ is the Archimedes force, $\boldsymbol{F_{ADw}}$ is the active drag force in water and $\boldsymbol{F_{ADa}}$ is the active drag force in air. The first two forces give the emerged/submerged ratio, while the last two forces are responsible of the transport in the horizontal plane.

The equation (3) is solved on vertical and horizontal planes. In the vertical plane, the equation is the balance between weight

force and Archimede reaction and its solution corresponds just to use the reduced mass of the target.

On the horizontal plane we solve the equation distinguishing between laminar and turbulent boundary layer. The resistence to the motion for a laminar boundary layer is given only by the viscosity friction and so we solve the Stokes law:

$$\mu_w \cdot R_W \cdot (\boldsymbol{u_c} - \boldsymbol{u_B}) + \mu_A \cdot R_A \cdot (\boldsymbol{w} - \boldsymbol{u_B}) = 0 \tag{4}$$

where $\mu_w$ and $\mu_A$ are the water and air viscosity, respectively, $R_W$ and $R_A$ is the submerged and emerged radius, $\boldsymbol{u_c}$ and $\boldsymbol{w}$

are the current and wind velocity, while $\boldsymbol{u_B}$ is the object velocity.

If the boundary layer is turbulent the resistence to the motion is given by the balance of the kinetic energy exchange between the moving body and the fluids:

$$\frac{1}{2} \cdot C_{D_W} \cdot \rho_w \cdot S_S \cdot (\boldsymbol{u_c} - \boldsymbol{u_B})^2 + \frac{1}{2} \cdot C_{D_A} \cdot \rho_A \cdot S_E (\boldsymbol{w} - \boldsymbol{u_B})^2 = 0 \tag{5}$$

where $S_S$ and $S_E$ are the submerged and emerged area and $C_{D_W}$ and $C_{D_A}$ are the corresponding drag coefficients.

Finally we estimate the Probability of Containment (POC) overlapping a spatial grid on the geophaycal one; we set arbitrarly each grid box to 0.06x0.06° and we calculate the particles percentage for each grid box, assigning a color bar from blue to red to distinguish the corresponding POC values from a minimum to a maximum value. The probability cumulative value is



calculated summing the values of highest probability of the individual cells then it could correspond to cells not geographycally consecutive. A final diary is written to storage the vertices of the cells of high probability.

## 2.3 Numerical Experiments

In this work the LEEWAY model is used to reproduce a real event occurred in the western Thyrrenian Sea, along the coast
of Sardinia: the day July $11^{th}$ 2013 a man was seen for a last time at UTC $08:30pm$ (Start point) approximately on board of a ferry that was on duty from Cagliari (Sardinia Island) to Civitavecchia (Italy) and only shortly before midnight of the $12^{th}$ July 2013 (End point) the alert was given. His body was recovered at UTC $10:30am$ of $12^{th}$ July 2013 in a point of known coordinate (Tab. 1).

The LKP is a critical step, since the accuracy of this information is decisive for the outcome of the search. In this task, we
represent it as a line between start and end points (Tab. 1) recorded from the Automatic Information System data provided by General Headquarters of the Italian Coast Guard (Fig. 1).

Numerical simulations of PIW trajectories are done using two different methodologies. The first one consists in the $leeway$ calculation by means of the equations described in (1), (2a) and (2b), while in the second one the $almost$ deterministic laws described by equations (4) and (5) are used.

In both the methodologies we rely on the conclusions of Breivik and Allen (2008) about the forcings perturbation, then they are considered of secondary importance especially in our stable and relatively homogeneous conditions. We use the mean standard deviation of the current and wind components, calculated on the spatial grid and on the period of the simulation (43 hours): the estimate is $0.07\ m \cdot s^{-1}$ on current east component and $0.1\ m \cdot s^{-1}$ on the nord one, while it is $2.68\ m \cdot s^{-1}$ on wind east component and $2.93\ m \cdot s^{-1}$ on the nord one.

We execute three different experiments sets seeding a total of 1000 particles in 8 time steps from start to end point of Last Know Position and subsequently estimating the smallest distance of the mass center of each subset of particles from the recover point. The first set of experiments is a study of sensitivity about the jibing value, which is changed from 1% to 8% for each of all the tabled positions (sitting, vertical, horizontal/survival, horizontal/deceased and $unknow$ status).

The second set of experiments is executed after choosing the best solution from the precedent study: in the regression line used
to calculate the downwind and crosswind $leeway$ components we check different coefficients of the time-invariant Gaussian perturbation ($\epsilon_n$) which Breivik and Allen (2008) introduced to include the variance increasing with wind speed.

Finally we execute the third set of experiments: now the solutions are calculated using the $almost$ deterministic approach. The human body geometry is assimilated to a cylinder with a ratio height/width between 4 and 7, according to Hoerner (1965), and then the projected frontal area, that is responsable predominantly of the drag, is calculated. Due to air in the lungs we suppose
that the body is submerged in standing position until to thorax and then we calculate the submerged and emerged part. To include the error about the calculation of the real surface exposed to the flow, we introduce two different perturbations: a vertical random oscillation between the thorax and the neck and a random rotation around the vertical axis; at the same time we assume a nonsignificant Stokes drift. Many experiments are executed to find the critic Reynolds number which marks the transition from laminar to turbulent boundary layer. The critic values between 120000 and 170000 are checked and in correspondence





different runs are produced using drag coefficients between 0.98 and 1.2, estimated according to Zdravkovich (1997) curves. Finally we find the critic Reynolds number and the drag coefficients corresponding to the solution where the seeded particles subsets distance from the rescue point/time is the smallest and use these values to perform a test using a dummy seeded in a different geophyc area, the Sicily Channel. The data about latter experiment are in the Tab. 3 and 4. Now the target is geo-

5 metrically scaled and the statistic of the new forcings is included: the estimation is a standard deviation 0.09 $m \cdot s^{-1}$ in current east component and 0.11 $m \cdot s^{-1}$ in the nord one, while it is 4.07 $m \cdot s^{-1}$ in wind east component and 4.25 $m \cdot s^{-1}$ in the nord one.

## 3 Results

In this section the principal results of each sets of experiments are showed. The first set of experiments is the study of sensitivity

on the jibing value for all the tabled positions. Here the $unknown$ status, corresponding to consider high error variances in the statistical calculation of the $leeway$, gives the best results; the different jibing checked values don't change significatively the results then we opt for the standard value 4%. The final map is visible in Fig. 2 where it is showed that at the rescue time the target is outside of the particles cloud. The distance of mass center of each seeding group from the rescue point/time is visible in Fig. 3: we verify that the $3^{rd}$ group, seeded between the 9:30am and 10:00am, is the closer one to the target, with a distance

which oscillates around the value of 9 km; the smallest absolute distance (about 2 $km$) is reached 31.5 hours later respect to the rescue time and it corresponds to the $4^{rd}$ group one, seeded between the 10:00am and 10:30am.

The second set of experiments is to study the influence of the perturbation coefficients on the results. It is helpful to remember that in the $leeway$ calculation by means of the regression line, they are introduced to include the variance increasing with the wind speed. In our experiments we verify that the spread is such that the particles final cloud includes the target at rescue time

(Fig. 4) when the slope is $a_n$=a+$\epsilon_n$/5 and the offset is $b_n$=b+2 $\cdot \epsilon_n$, i.e. the perturbation coefficients are 4 times greater that ones proposed in Breivik and Allen (2008) . The mass center distance of the $3^{rd}$ group is again the smallest one and it oscillates around the value of 5.5 $km$ (Fig. 5); the smallest absolute distance (about 4.3 km) is reached 5 hours later respect to the rescue time and it corresponds always to the $3^{rd}$ group one. In this experiment then the uncertainty about the time of the accident is reduced and the distance of the mass center of the favoured seeded group from the rescue point is almost halved but the

dispersion is now very higher.

To reduce the dispersion and to give a greater weight to the advection, a different option about the perturbation coefficients is checked: if wind velocity is lower than 10 $m \cdot s^{-1}$ we increase only the offset perturbation coefficient while for wind velocity greater that 10 $m \cdot s^{-1}$ we increase only the slope one, according to different weight of the offset/slope of the regression line at lower/higher wind speeds . Now the particles cloud includes the target at the rescue time and the dispersion of the particles

is reduced (Fig. 6) but the distance at the rescue time of the mass center of the $3^{rd}$ group from the rescue point is very similar to that one in the first experiment resulting be about 8.9 $km$ (Fig. 7). Again, as in the first experiment, the smallest absolute distance (about 2.8 $km$) is reached by $4^{rd}$ seeded group 31.5 hours later respect to the rescue time.

The precedent experiments pointed out the different performances of the model when the real statistic on forcings and the





statistical regression line to calculate the *leeway* velocity are used. Not only the heteroshedasticity of the experimental dataset compiled by Allen and Plourde (1999) needs to be correctly accounted for to optimize the model performances, but we need also to choose the best perturbation coefficients to use in the regression line. At now we don't know them in advance and we only can say that they should be such that the dispersion doesn't swamp the advection role and at the same time it should

induce to maximize the POC.

The third set of experiments consists in to simulate the PIW drift by means of the *almost* deterministic law. After checking the projected frontal area by means of the allometric parameters for a "standard human body", according to Herman (2006), we choose a ratio height/width equal to 4.44. The final critic Reynolds number range is estimated between 165000 and 168000. The Reynolds numbers height based are always between $O(10^4)$ and $O(10^5)$; due to these values and to roughness role on the

boundary layer modifications (Yeo and Jones, 2011) we conclude that the PIW boundary layer in our experiment is often laminar with transition to turbulence prevalently precritic. That corresponds to say that the drag crisis never occurs, the boundary layer never separates and the drag coefficient in the turbulent boundary layer is constant for a large Reynolds numbers range. In our experiments the drag coefficients corresponding to the best results is 1.12. All the parameters to solve the forces balance equation are in the Tab. 2.

The results now show that at rescue time the cloud includes the target (Fig. 8) and at same time the dispersion of the particles looks low and qualitatively comparable with that one in Fig. 6 where the *leeway* perturbation coefficients have been setted separately on the slope or on the offset of the regression line according to wind velocity; the absolute distance of $3^{rd}$ group of particles from the rescue point is minimum (Fig. 9) and it is about 1.3 $km$.

Finally we checked this approach in a different area (Sicily Channel), using experimental data (Tab. 3) and scaling geometri-
cally our target (Tab. 4). The results show that the particles cloud includes the target and the minimum distance of the particles cloud is reached at the rescue time (Fig. 10), resulting about 5.8 $km$. The target is located on the external border of the cloud (Fig. 11), then the POC is from medium to low value; it seems that approximation on the POC calculation is greater than that one estimated in the Sardinia area but we believe that the POC can be improved if the set of particles is clustered according to the hydrodynamic structures and not only according to seeding time. Actually we are working to this hypothesis.

**4   Conclusions**

We have presented the results of numerical experiments performed to estimate the Probability of Containment (POC) of Person in Water (PIW). We have referred to a real event occurred in the western Thyrrenian sea. The LEEWAY model, nested with the sub-regional hydrodynamic model TSCRM has been used. The real statistic of the forcings on the observation period was accounted for and the Last Known Position has been based on the trajectory of the ferry, recorded by the Automatic Information

System, during 3.5 hours; to reproduce the event a set of 1000 particles seeded in 8 time step has been used.

The objective of the work has been to validate an *almost* deterministic law to calculate the *leeway*, based on the boundary layer theory; these results have been compared with those obtained estimating the *leeway* by means of the standard statistical approach. The latter has produced the best results when the perturbation coefficients in the regression line have been changed





separately on the offset or on the slope of the regression line according to wind velocity.

To calculate the $leeway$ in the $almost$ deterministic approach the PIW has been schematized as a cylynder; the critic Reynolds number has been found in the range between 165000 and 168000 and the drag coefficients have been estimated equal to 1.12. The results now have showed that at the rescue time the particles cloud includes the target and the $3^{rd}$ subset of particles is

representative of the event then we also can suppose the time of the accident with good approsimation. The results estimated by means of the standard statistical approach are similar but with the new approach the distance of the mass center of the favoured subset of particles is strongly reduced 6.5 times, resulting of $1.3km$. The tests in Sicily Channel using a similar target have confirmed the goodness of the method but we believe that the results can be further improved subdividing the set of particles according not only to seeding time but also to hydrodynamic structures; we are actually working to this hypothesis.

The method needs to be improved and the first step will be including the Stokes drift; anyway the results here obtained encourage to explore other different target categories. We believe that on the long time it could permit to overcame the limitations connected with the standard approach based on the tabled statistical parameters providing leeway-drift formulas for practical use based on appropriate Reynolds critic numbers and drag coefficients for specific targets. This idea is supported by the realistic option to execute tests in naval tank, simulating different meteomarine conditions, scaling opportunely specific targets

and then overcoming the costs and difficulties to acquire field data.

*Acknowledgements.* This paper was supported by the Italian Project PON-TESSA (PON01-02823) founded by the Italian Ministry for Research - MIUR. We would like to thank to C.te Sirio Faé of Italian Coast Guard for providing the data about our experiments and for supporting in the interpretation of the IAMSAR Manual. Finally thanks to Ing. Massimo Miozzi of the CNR-INSEAN for his constructive comments during the work.



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

**Figure 1.** The Last Known Position, represented by the blue particles distribution at the alert time; it is based on the AIS system that gives the ferry position every 6 seconds (green line on the left of the picture). The coordinates of start and end point are defined in Tab. 1.

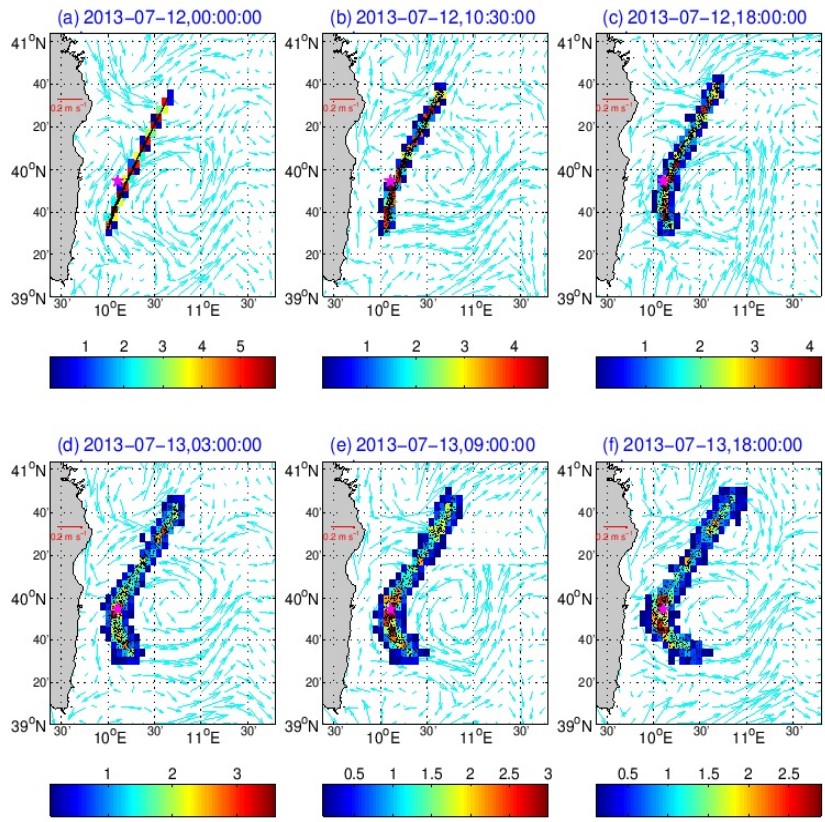

**Figure 2.** Particles cloud estimated by the LEEWAY model overlapped to the hydrodynamic field. Here the *leeway* comes by statistical law as in Breivik and Allen (2008). The probability of containment (POC) also is computed and it varies from blue/minimum to red/maximum value in each grid box of 0.6x0.6°. At rescue time (subplot (b) in the panel), the target (magenta point) is outside the particles cloud; it will be included in the cloud only 7.5 hours late (subplot (c) in the panel).




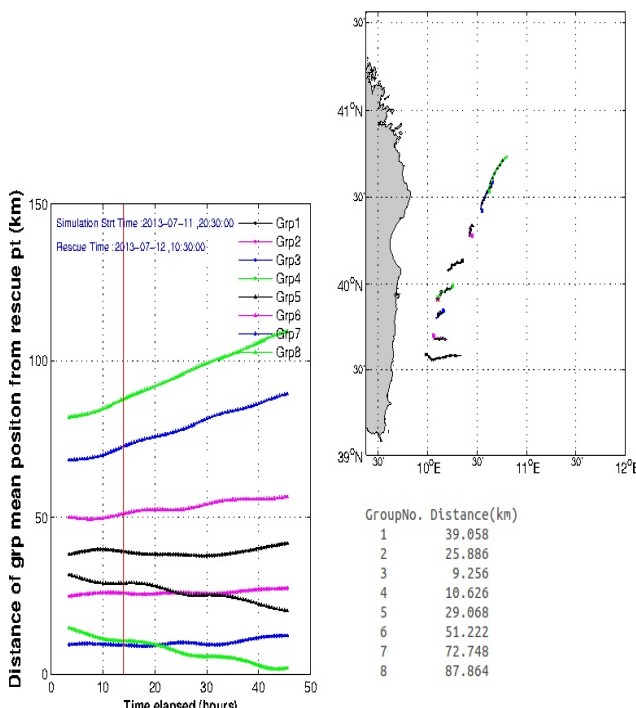

**Figure 3.** Distance of the mass center of each group seeded every 30 minutes from 8:30$pm$ (Group 1) of the 11 july 2013 to next midnight (Group 8). The vertical red line marks the rescue time. Here the $leeway$ is calculated by means of the statistical law as in Breivik and Allen (2008). May be possible verify that the lower distances at the rescue time (values at right and low part of the panel) is reached by the mass center of the $3^{rd}$ subset of particles (seeded from 9:30$pm$ to 10:00$pm$). The smallest distance is reached 31.5 hours later by the $4^{th}$ subset (seeded from 10:00$pm$ to 10:30$pm$) and it is about 2 km. On the upper right side of the picture the trajectories of the each group mass center during all the simulation time (43 hours) are visible.




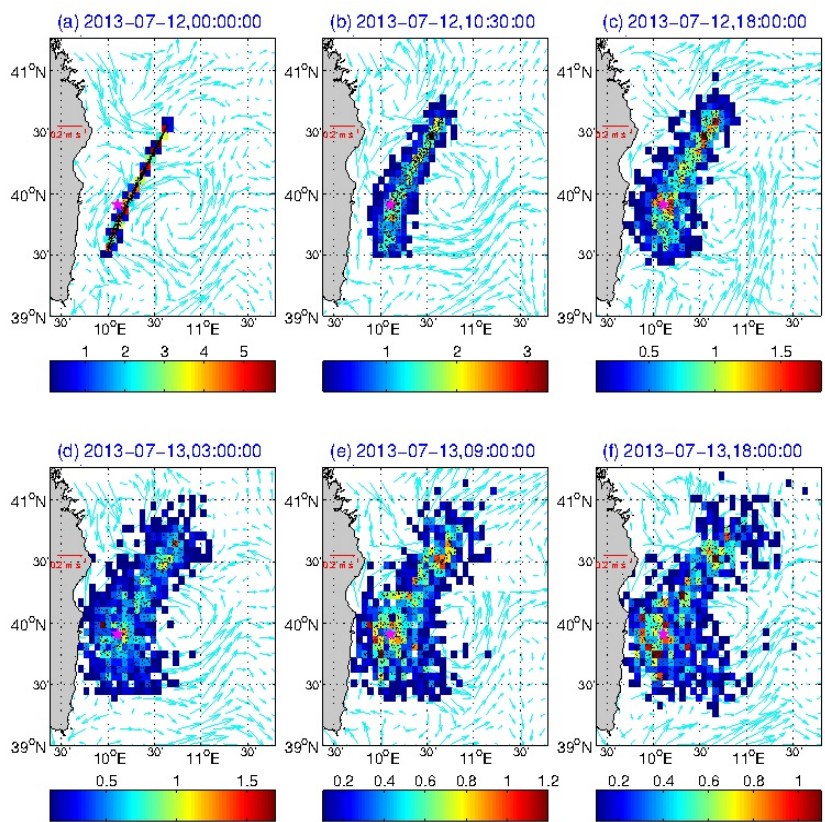

**Figure 4.** Here the perturbation coefficient is 4 time greater that one in Breivik and Allen (2008); in particular a common perturbation coefficient was setted at the same time either in slope and in the offset of the regression line. Now the particles cloud includes the target at the rescue time but the dispersion is much larger than that one in figure 2.




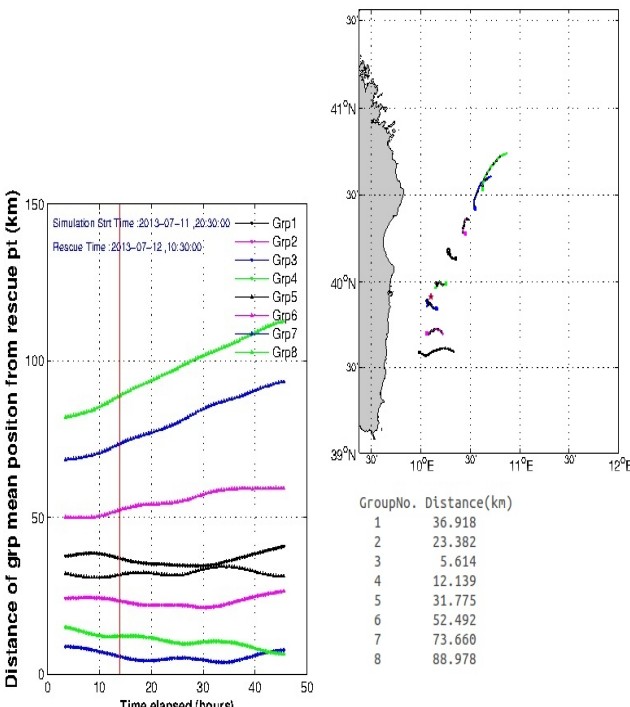

**Figure 5.** Distance of the mass center of each group seeded every 30 minutes from 8:30$pm$ (Group 1) of the 11 july 2013 to next midnight (Group 8). The $leeway$ perturbation coefficient was setted at the same time on the offset and on the slope of the regression line as 4 times that ones corresponding to the results in figure 3. Now the mass center of $3^{rd}$ subset (seeded from 9:30$pm$ to 10:00$pm$) has the smallest distance at the rescue time and it is about 5.6 $km$. The smallest absolute distance is about 4.3 $km$ and it is reached 5 hours later by the same subset of particles. The trajectories of the each group mass center during all the simulation seem slowly changed respect to the precedent experiment.





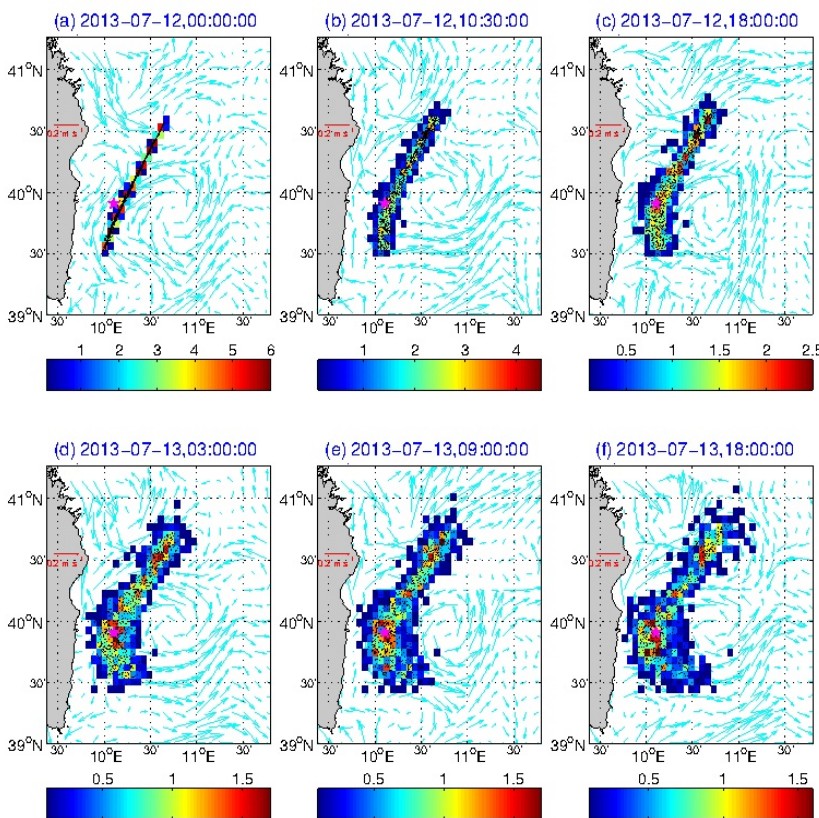

**Figure 6.** Here the common perturbation coefficient was setted separately on the offset or on the slope of the regression line according to the wind velocity: for wind velocity lower than $10\ m \cdot s^{-1}$ the *leeway* perturbation coefficient is 4 time that one in Breivik and Allen (2008) only on the offset of the regression line while for wind velocity greater that $10\ m \cdot s^{-1}$ only that one of the slope was changed. The particles cloud includes the target at the rescue time (panel (b) in the figure) and the dispersion is reduced respect to that one in the precedent experiment.





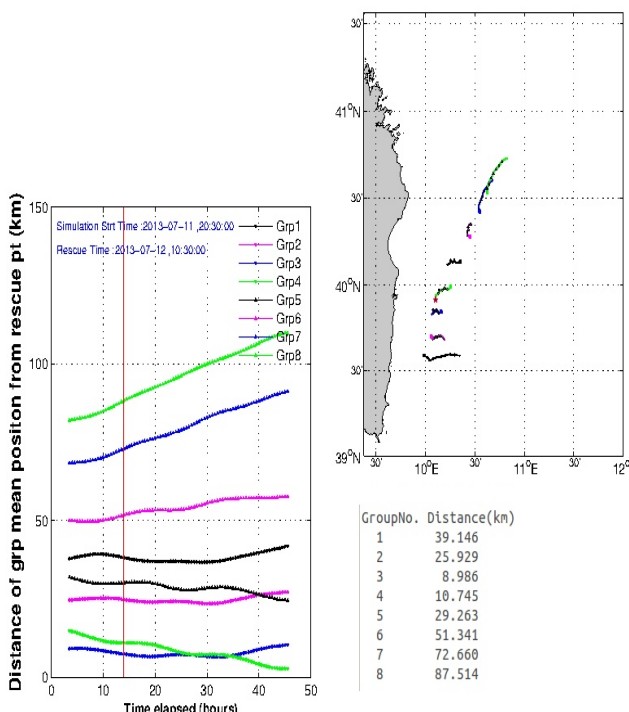

**Figure 7.** Distance of the mass center of each group seeded every 30 minutes from 8:30$pm$ (Group 1) of the 11 july 2013 to next midnight (Group 8). The $leeway$ perturbation coefficient is setted separately on the offset or on the slope of the regression line according to wind velocity. Now the mass center of the $3^{rd}$ subset of particles (seeded from 9:30$pm$ to 10:00$pm$) has the smallest distance at the rescue time: it is about 8.9 $km$ and it is very similar to that one in the first experiment . The smallest absolute distance is about 2.8 $km$, estimated 31.5 hours later and it belongs to the mass center of the $4^{rd}$ group.



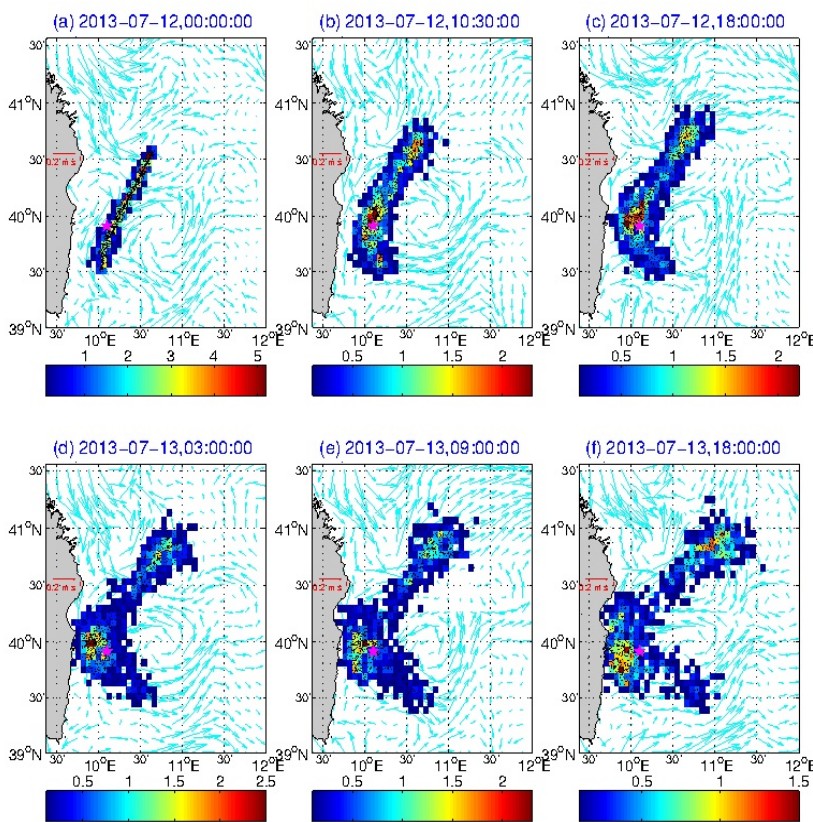

**Figure 8.** Particles cloud simulated by the model overlapped to the hydrodynamic structures. Here the *leeway* is calculated by means of the forces balance equation. The cloud includes the target at the rescue time and the dispersion seems not to swamp the advection; in particular the particles cloud seems to separate on a large time according to hydrodynamic structures (subplot (e) and (f) in the panel).



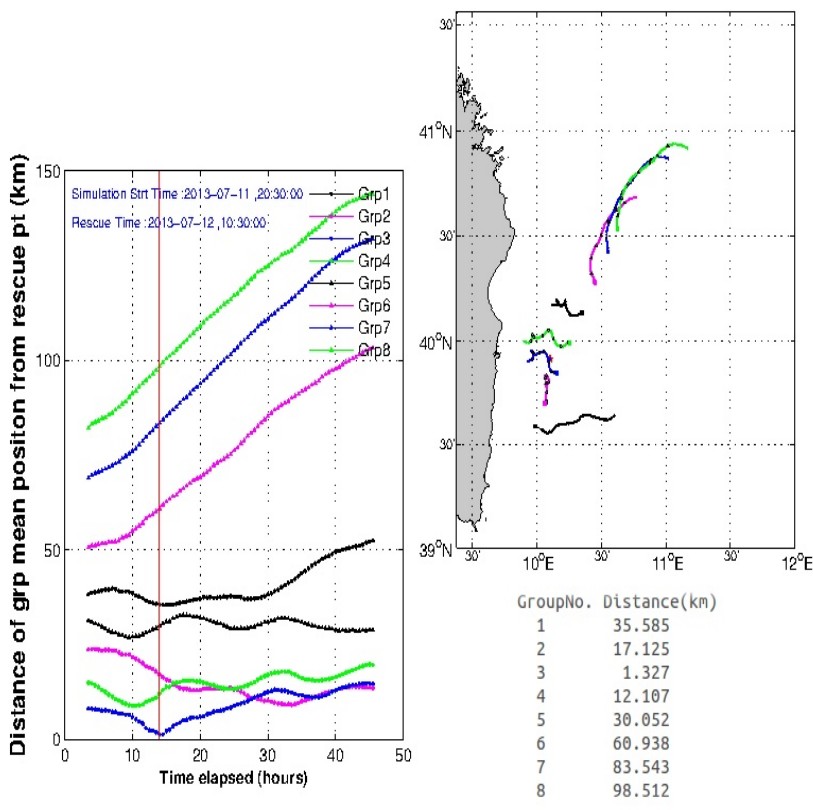

**Figure 9.** Distance of each seeded group mass center from the rescue time/space. The seeding was every 30 minutes from $8:30pm$ (Group 1) to midnight (Group 8). Here the $leeway$ is calculated by means of the forces balance equation. May be possible verify that now the solution is better than the previous ones: at the rescue time the $3^{rd}$ group (seeded from $09:30pm$ to $10:00pm$) has the smallest absolute distance and it is about $1.3\ km$.




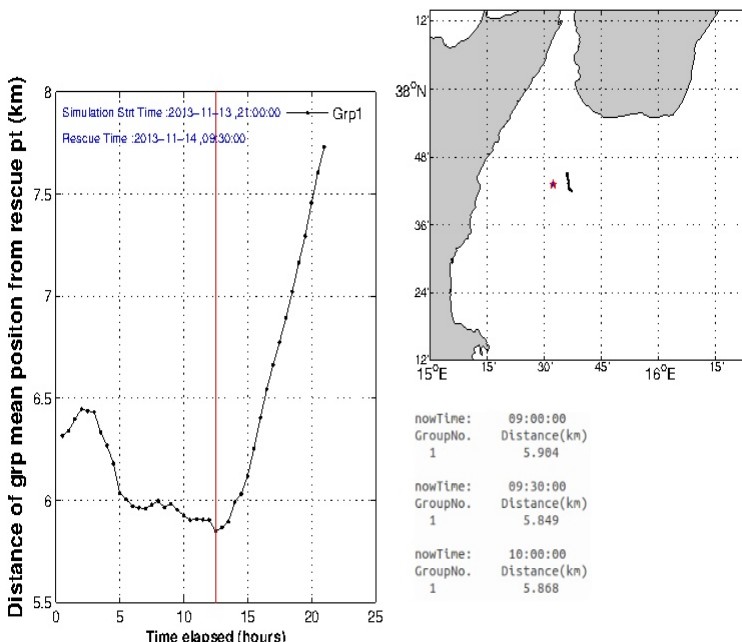

**Figure 10.** Distance of the mass center of the particles set seeded in a known time/point to simulate the trajectory of a dummy in the Sicily Channel. Again the *leeway* is calculated by means of the forces balance equation. On the upper right side of the picture the trajectory of the mass center during all the period of the simulation (21 hours) is visible.



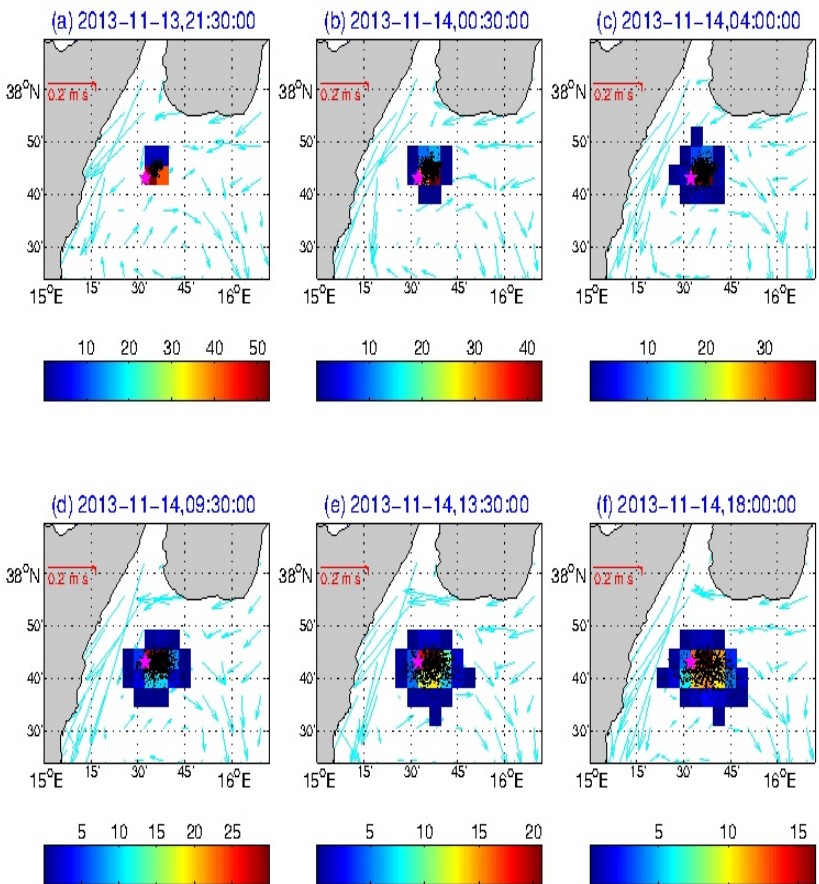

**Figure 11.** Particles cloud overlapped on the hydrodynamic field in the experiment executed in the Sicily Channel. In the panel (d) the configuration at the rescue time is visible. Far from the coast the current is prevalently easward but the particles subset influenced by the current near the coast well simulate the trajectory from the seeding point to the rescue point/time.





**Table 1.** Data inherent the incident

|  | $\phi$ | $\lambda$ | Date (mm-dd-yyyy) | Time (UTC) |
|---|---|---|---|---|
| Start Point | 39° 33.16′N | 009° 57.99′E | 07-11-2013 | 08:30 pm |
| End point | 40° 34.04′N | 010° 38.99′E | 07-12-2013 | 00:00 |
| Recovered point | 39° 54.71′N | 010° 06.29′E | 07-12-2013 | 10:30 am |

**Table 2.** Parameters used to solve the forces balance equation about the Person In Water

| | |
|---|---|
| Water Density $(kg/m^3)$ | 1027.0 |
| Water cinematic viscosity $(m^2/sec)$ | $0.974 \cdot 10^{-6}$ |
| Air Density (kg/m3) | 1.2 |
| Air cinematic viscosity $(m^2/sec)$ at $20^oC$ and $1\ atm$ | $1.41 \cdot 10^{-5}$ |
| Human body Density $(kg/m^3)$ | 985.0 |
| Standard Human Body weight $(kg)$ | 70 |
| Standard Human Body height $(m)$ | 1.7 |
| Critic Reynolds Number range | 165000-168000 |
| Water/Air Drag coefficient | 1.12 |

**Table 3.** Data about the second experiment. A dummy seeded in a known point and rescued after 12.5 hours

|  | $\phi$ | $\lambda$ | Date (mm-dd-yyyy) | Time (UTC) |
|---|---|---|---|---|
| Start/End Point | 37° 45.00′N | 015° 36.00′E | 11-13-2013 | 08:30 pm |
| Recovered point | 37° 43.00′N | 015° 32.32′E | 07-12-2013 | 09:30 am |

**Table 4.** Parameters to solve the force balance equation for the dummy in the Sicily Channel

| | |
|---|---|
| Water Density $(kg/m^3)$ | 1027.0 |
| Water cinematic viscosity $(m^2/sec)$ | $1.1 \cdot 10^{-6}$ |
| Air Density $(kg/m^3)$ | 1.2 |
| Air cinematic viscosity $(m^2/sec)$ at $10^oC$ and $1\ atm$ | $1.5 \cdot 10^{-5}$ |
| Dummy weight $(kg)$ | 50 |
| Dummy height $(m)$ | 1.5 |