# Peer review of "Evaluation of the Search and Rescue Leeway model into the Tyrrhenian sea: a new point of view"

_Natural Hazards and Earth System Sciences, 2016_

## Referee Comment (RC1) · Anonymous Referee #1 · 10 May 2016

General comments The work presented by the authors focuses on a crucial aspect of the search and rescue (SAR) operations, the assessment of the most probable search area by means of a stochastic model. The manuscript is well organized, with a clear description of the methods used; the results obtained may be useful for future applications. For this reasons I would recommend the acceptance of this contribution after some minor revisions and technical corrections. The reviewer also suggests publishing the manuscript after the revision of English language.

Specific comments The manuscript well describes the role of operational ocean forecasting system in the SAR operations (GODAE BLUElink used by the Australian Maritime Safety Authority and MFS for the Mediterranean Sea), and underlines that an efficient SAR planning requires high-quality environmental data with a particular emphasis on surface currents. Nevertheless, the authors never mention the use of real–time high-

resolution currents measured by coastal HF radar systems to backtrack drifting objects. For marine safety and SAR activities, the integration between HF radar currents and backdrift estimation technique is useful to locate the origin of a marine accident if a shipwreck appears in the sea or even in the search of the wreckage area in the case of an aircraft accident. I think the paper might read better after including this topic in the introduction paragraph (for references see O'Donnell et al., 2005; Abascal et al., 2012, Cianelli et al. 2015).

References: Abascal A.J., Castanedo S., Fernández V. and Medina R.: Backtracking drifting objects using surface currents from high-frequency (HF) radar technology, Ocean Dynamics 62:1073–1089, 2012.

Cianelli D., Iermano I., Mozzillo P., Uttieri M., Zambardino G., Buonocore B., Falco P., Zambianchi E.: Inshore/offshore water exchange in the Gulf of Naples, Journal of Marine Systems - 145, 37–52, 2015.

O'Donnell J, Ullman D, Spaulding M, Howlett E, Fake T, Hall P, Tatsu I, Edwards C, Anderson E, McClay T, Kohut J, Allen A, Lester S, Lewandowski M.: Integration of Coastal Ocean Dynamics Application Radar (CODAR) and Short-Term Prediction System (STPS) surface current estimates into the Search and Rescue Optimal Planning System (SAROPS). US Coast Guard Tech.Rep., DTCG39-00-D-R00008/HSCG32-04-J-100052, 2005.

How was the study area for dummy test experiment selected? The authors should clarify this point in the "Numerical Experiments" section.

In order to improve the paragraph "Results", the authors should comment the obtained outcomes in the frame of the surface dynamics features acting in the selected area at the time of the experiment.

Technical corrections

Page 1, line 6: "Thyrrenian"> "Tyrrhenian"

Page 1, line 9: "Thyrrenian"> "Tyrrhenian"

Page 1, line 16-19: [Meteocean and environmental forecast are………. the effectiveness of the search operation (Breivik et al., 2013).] rewrite the sentence after the revision of English language.

Page 2, line 16: […regional operational oceanography systems following on from of the successful…] remove "on" and "of".

Page 2, line 30-31:[…the operational oceanography service in Southern Italy and to integrate this service…] rewrite the sentence replacing "service" with a synonymous.

Page 2, line 33: "minister of the infrastructures and transport" > "Ministry of Infrastructures and Transport". Page 3, line 12: "fo"> "of"

Page 3, line 13: remove "executed".

Page 3, line18-19: […conservation of temperature, salinity and assumes…] rewrite as […conservation of temperature and salinity, the model assumes….]

Page 3, line 20: " by and adaptation" remove "and".

Page 3, line 30-32: [Surface momentum and buoyancy ….. provided by the European Centre for Medium Range…] rewrite the sentence after the revision of English language.

Page 4, line 10: "numerical models of measurements" rewrite as "numerical models".

Page 6, line 4: "Thyrrenian"> "Tyrrhenian"

Page 6, line 22-23: [The first set of experiments …. and unknow status)] rewrite the sentence providing more detail about the estimation of the jibing value.

Page 8, line 1: "heteroshedasticity"> "heteroscedasticity".

Page 8, line 13:" drag coefficients"> "drag coefficient".

Page 8, line 31: "bounday"> "boundary".

Page 9, line 11-13: [We believe that ......and drag coefficients for specific targets.] rewrite the sentence after the revision of English language.

References

Page 10, line 6-7: "Bell, M., Schiller, A., Traon, P.-Y. L., Smith, N., Dombrowsky, E., and Wilmer-Becker, K.: GODAE: The Global Ocean Data Assimilation Experiment Forecasting, Oceanography, 22, 2009."

Rewrite as

"Bell, M.J., Lefèbvre M., Le Traon P.-Y., Smith N.and K. Wilmer-Becker. GODAE: The Global Ocean Data Assimilation Experiment. Oceanography 22(3):14–21, 2009.

Page 10, line 8-9: "Bell, M., Tran, P.-Y. L., Lefebvre, M., Smith, N., and Wilmer-Becker., K.: An introduction to GODAE OceanView, Journal of Operational Oceanography, 8, 2–11, 2015."

Rewrite as

"Bell M.J., Schiller A., Le Traon P.-Y., Smith N.R., Dombrowsky E. and K. Wilmer-Becker An introduction to GODAE OceanView, Journal of Operational Oceanography,8(1), 2-11, 2015."

Page 10, line 12-13: "Brassington, G., Pugh, T., abd E. Schulz, C. S., Beggs, H., Schiller, A., and Oke, P.: BLUElink: Development of operational oceanography and servicing in Australia., Journal of Research and Practice in Information Technology, 39, 151–164, 2007."

Rewrite as

"Brassington, G., Pugh, T., Spillman C., Schulz E., Beggs, H., Schiller, A., and Oke, P.: BLUElink: Development of operational oceanography and servicing in Australia.,

Journal of Research and Practice in Information Technology, 39, 151–162, 2007."

Page 10, line 30: "Mellor, G.: An equation of state for numerical models of oceans and estuaries, J. Atmos. Oceanic Tech., 8, 8609–611, 1991."

Rewrite as

"Mellor, G.: An equation of state for numerical models of oceans and estuaries, J. Atmos. Oceanic Tech., 8, (4), 609-611, 1991."

Page 10, line 31: "Mellor, G. and Yamada, T.: Development of a turbulence closure model for geophysical fluid problems, Rev. Geophys. Space Phys, 20," delete "Space Phys"

Page 10, line 35-36: "Pinardi, N., Allen, I., Mey, P. D., Korres, G., Lascaratos, A., Traon, P. L., Maillard, C., Manzella, G., and Tziavos, C.: The Mediterranean ocean Forecasting System: first phase of implementation (1998-2001)., Annales Geophysicae, 21, 3–421, 2003"

Rewrite as

Pinardi, N., Allen, I., Demirov, E., De Mey, P., Korres, G., Lascaratos, A., Le Traon, P.-Y., Maillard, C., Manzella, G., and Tziavos, C.: The Mediterranean ocean Forecasting System: first phase of implementation (1998–2001), Annales Geophysicae 21, 3-20, 2003.

Page 10 line 37: check the reference "Richardson: Drifting in the wind, Applied Ocean Research, 15, 121–135, 1993."

Page 11, line 6: "insigh" > "insight".

Page 11, line 10-11: insert the paper volume and page numbers.

Figure 1 - Caption: define the red and green point on the right panel of the figure.

---

## Author Comment (AC1) · 15 Jun 2016

The authors should thank the referee #1 for his constructive comments. We agree with him about the opportunity of improving the Results discussion, commenting the outcomes in the frame of surface dynamic structures. This part of the discussion, infact, is useful both to highlight the advection role in the particles dispersion and to improve the calculation of the probability of containment (POC). We are working, infact, to delimite the principal hydrodynamci strucutres in the seeding area so as to correlate them quantitatively with the particles distribution. This part of the research is not included here because it needs of insights that are being; we hope to show interesting results in a next paper. The strong correlation between the surface dynamics and the particles distribution besides underlines the importance to have an efficient operational ocean prediction system for the SAR activities. A 2nd (but not for its importance) improvment

of the paper, following the comments of the referee, consists in the discussion about the use of the HF radar for marine safety and SAR activities. The HF radar, infact, allows both to measure the surface currents on real time/high resolution and to locate the origine of the accident; for this reason we think it should be included in an coastal integrated system for SAR activities.

The pdf supplement file added here is the new version of the manuscript where the improvements proposed by the referee #1 and two additional figures are included.

Please also note the supplement to this comment:
http://www.nat-hazards-earth-syst-sci-discuss.net/nhess-2016-109/nhess-2016-109-AC1-supplement.pdf

**Supplement:**

**Evaluation of the Search and Rescue Leeway model into the Tyrrhenian sea: a new point of view**

Antonia Di Maio[1], Mathew V. Martin[2], and Roberto Sorgente[3]

[1]Marine Technology Research Institute INSEAN, National Research Council of Italy, Rome, Italy
[2]Centre for Oceans, Rivers, Atmosphere and Land Sciences, Indian Institute of Technology Kharagpur, West Bengal, India
[3]Institute for Coastal Marine Environment IAMC, National Research Council of Italy, Oristano, Italy

*Correspondence to:* Antonia Di Maio (antonia.dimaio@cnr.it)

**Abstract.** The trajectories prediction of the floating objects above the sea surface represents an important task in the search and rescue (SAR) operations. In this paper we show how it can be possible to estimate the most probable search area by means of a stocastic model, schematizing appropriately the shape of the object and evaluating the forces acting on it. The LEEWAY model, a Montecarlo-based ensemble trajectory model, has been used: here both the statistical law to calculate the *leeway* and an *almost* deterministic law inspired by the boundary layer theory have been considered. The model is nested with the sub-regional hydrodynamic model TSCRM (Tyrrhenian Sicily Channel Regional Model) developed in the framework of PON-TESSA (National Operative Programs-TEchnology for the Situational Sea Awareness) project. The main objective of the work is to validate the new approach of *leeway* calculation relying on a real event of Person in Water (PIW), occurred in the Tyrrhenian Sea in July 2013 . The results show that by assimilating a human body to a cylinder and estimating both the transition from laminar to turbulent boundary layer and the drag coefficients, it can be possible to solve a forces balance equation which allows to estimate the search area with good approximation. This new point of view leads to the possibility to check the same approach also on other different categories of targets, so as to overcome in the future the limitations associated with calculation of *leeway* by means of the standard statistical law.

**Keywords.** LEEWAY Model, *leeway*, Person In Water, Search and Rescue

**1 Introduction**

Meteocean and environmental forecast are being increasingly used in operational decision making in the sea for demographic, geographic and strategic applications. Safety and assets at sea are a shared objective of many countries: having an efficient ocean forecast system is essential to improve prediction of sea and to provide useful environmental ocean information so as to increase the effectiveness of search operations (Breivik et al., 2013). In the event of accident, timely Search and Rescue (SAR) intervention is helpful in significantly lowering the loss of life and also to contain the damage. A considerable amount of resources is currently invested in maintaining SAR capabilities by major maritime nations. However, on many occasions, the available SAR capabilities prove to be inadequate to provide timely assistance in distress scenarios. Nevertheless, the effectiveness of the available SAR capabilities can be increased if we exploit the high-quality environmental forecast data for

SAR planning. An example of operational ocean forecasting and services coupled with Search and Rescue (SAR) activities are represented by GODAE BLUElink operational ocean prediction system (Brassington et al., 2007), used by the Australian Maritime Safety Authority. A list of global and regional operational ocean forecasting systems, supported by the Global Ocean Data Assimilation Experiments - GODAE (Bell et al., 2009), can be found in Davidson et al. (2009), while a recently review of the evolution of the global and regional forecasting system from GODAE into GODAE OceanView (Bell et al., 2015) is described in Tonani et al. (2015). These systems are based on ocean general circulation models (OGCM) and data assimilation techniques that are able to correct the model with information inferred from different types of observations, acquired by various sensors and platforms.

In the coastal areas, the high-frequency radars are crucial to acquire the surface two-dimensional current data (Barrick et al. (1977); Barrick et al. (2012); Cianelli et al. (2015); Paduan and Rosenfeld (1996)). Maps acquired by a coastal HF radar have been used both to obtain backtracked trajectories of floating object (Abascal et al. (2012); Abascal et al. (2009b); Berta et al. (2014)) and for SAR applications (Ullman et al. (2006); Ullman et al. (2003)). Some european countries performed an important effort toward the implementation of national HF radar networks (Quentin et al., 2013); (Carrara et al., 2014), but at present just one unified HF coastal radar network is started in the Mediterranean Sea. It is the Tracking Oil Spill and Coastal Awareness (TOSCA) network, covering the Aegean, Adriatic, Tyrrhenian, Ligurian and Balearic Seas; it started as support for decision-making process relative to marine accidents concerning oil spill pollution and search and rescue (SAR) operations (Bellomo et al., 2015). TOSCA network currently covers sensitive and environmentally relevant areas, affected by intense ship traffic and/or the presence of oil pipelines; nevertheless it is only the first step towards the building of an integrated HF radar system in the Mediterranean regional alliance and its coverage is relatively small yet, so it is not always involved in the ocean forecasting systems nor in the connecting application such as the SAR planning.

The first step in marine search planning is to determine the most probable area containing the searched object and that is even more important if the target is person in water (PIW). The definition of the probable search area is essentially linked to the quantification of some unknowns, such as the Last Known Position (LKP) and typology, shape and dimensions of the objects. Moreover, a PIW or object without propulsion is also subject to drift from ocean currents, wave action and direct wind action (Davidson et al., 2009). Hence, an effective SAR planning requires accurate environmental forecasting, especially regarding surface currents, temperature and surface winds for short range time-scale. Today, such information can be provided in near real time (NRT) from global and regional operational oceanography systems following from the successful implementation of the Global Ocean Observing System (GOOS).

In the Mediterranean Sea, operational forecast starting from the year 2000 are provided by the Mediterranean Forecasting System - MFS (Tonani et al., 2008). The MFS uses a horizontal grid of resolution of $1/16°$ and provides detailed forecast at a regional scale for the whole Mediterranean basin. The fundamental part of the system is the assimilation scheme for the blending of the observations into the model in order to provide the most accurate description of the past and the best initial condition for the forecast fields at large scale (Oddo et al., 2009).

In many cases, SAR activities may also require fields of interest on a high spatial resolution. Sub-regional forecast systems that can resolve small-scale processes as well as fronts characteristics of the study area are necessary. A numerical technique widely

used in operational oceanography to increase the horizontal resolution is the downscaling procedure, which permits to adapt the large scale features to the coastal areas by means of model domains cascade embedded (Pinardi et al., 2003). This has been achieved through implementation of a nested high resolution ocean model for the central Mediterranean basin developed in the framework of the project Development of TEchnology for Situational Sea Awareness (TESSA). This hydrodynamic model, named Tyrrhenian and Sicily Channel sub-Regional Model (TSCRM), is based on POM (Blumberg and Mellor, 1987) and has a horizontal resolution of 1/48°, equal to three time MFS.

TESSA project is supported by PON "Ricerca e Competivita' 2007-2013", which is a National Operational Programme of the Ministry for Education, University and Research of Italy. The general aim is to improve the products and services of the operational oceanography in Southern Italy and to integrate them with technological platforms. The latter are set up to disseminate information for the Situational Sea Awareness (SSA) also in support to the SAR activities at sea. In Italy these activities are performed by the Coast Guard, within the competence of the Ministry of Infrastructures and Transport of Italy.

In this paper we present some numerical results about the prediction of PIW trajectories in the central Tyrrhenian Sea. We use two different approaches for the $leeway$ calculation: the first approach is standard and it consists in the $leeway$ calculation by means of the statistical parameters tabled by Allen and Plourde (1999). The second one is a variant of the first approach: it consists in the $leeway$ calculation by means of the forces balance equation based on modifications of the target boundary layer. The main of this work is to validate the latter approach. We hope that in the near future the new method will permit to define a practical formula of $leeway$ calculation helpful not only in similar PIW cases but also in other targest cathegories.

The new approach is embedded in the LEEWAY model, a drift model based on a stochastic approach (Breivik and Allen, 2008), combined with environmental forecast data from TSCRM and European Centre for Medium Whether Forecast (ECMWF). The ensuing sections of the paper are organized as follows: section 2 describes the models used and the numerical experiments; an analysis of the model results is presented in section 3 together with possible future extensions. A summary and concluding remarks are given in section 4.

**2    Material and Methods**

[revised manuscript text omitted]

10    speed and divergence angle, gives a measure of how much the wind directly pushes on a object floating at the sea.

It is necessary to assume that the empirical coefficients of linear regression of the equations 1 and (2a) and (2b) also include the contribution of the Stokes drift (Breivik and Allen, 2008). Following this we assume that *leeway* can be only expressed as a function of the wind and we use a practical definition of *leeway* which doesn't tell the wind and wave influence apart. This approximation can be also extend to other small objects and to most sea states, according to Breivik et al. (2012).

[revised manuscript text omitted]

We perform three different sets of experiments seeding a total of 1000 particles in 8 time steps from start to end point of Last

Know Position and subsequently estimating the smallest distance between the mass center of each subset of particles and the retrieval point.

The first set of experiments is a check about the target tabled configurations (sitting, vertical, horizontal/survival, horizontal/deceased and *unknow* status) as well as a study of sensitivity about the change of the drift from a persistent direction to the opposite one (jibing) relative to downwind. This change will result in a sign change on the crosswind component. The jibing is set in the model as a frequency fixed on all the simulation time. Such information is very important: if the object doesn't jibe, the initial probability distribution could rapidly split in two equal probability distant areas, according to the initial uncertainty about the tack relative to downwind. On the contrary, the more frequent is the jibing, the more central is the distribution (Allen et al., 2010).

We check a jibing frequency from 1% to 8% for each tabled position.

The second set of experiments is performed after choosing the best solution from the previous study: in the regression line used to calculate the downwind and crosswind *leeway* components, we check different coefficients of the time-invariant Gaussian perturbation ($\epsilon_n$) which Breivik and Allen (2008) introduced to include the variance increasing with wind speed.

Finally we carry out the third set of experiments: here the solutions are calculated using the *almost* deterministic approach.

The human body geometry is assimilated to a cylinder with a ratio height/width between 4 and 7, according to Hoerner (1965); then the projected frontal area, that is predominantly responsable for the drag, is calculated. Due to air in the lungs we suppose that the body is submerged in standing position up to thorax and then we calculate both the submerged and emerged part. To include the error about the calculation of the real surface exposed to the flow, we introduce two different perturbations: a vertical random oscillation between the thorax and the neck and a random rotation around the vertical axis; at the same time we assume a nonsignificant Stokes drift. Many experiments are performed to find the critic Reynolds number which marks the transition from laminar to turbulent boundary layer. Each critic values between 120000 and 170000 is coupled with all drag coefficients between 0.98 and 1.12 (the latter estimated according to Zdravkovich (1997) curves) resulting in different runs. At last we find the critic Reynolds number and the drag coefficient corresponding to the solution where the distance between the seeded particles subsets and the retrieval point/time is the shortest.

To check the reliability of the *almost* deterministic approach we run the model in a different geographycal area, using a target *similar* to a person and referring to the critic Reynolds number vs drag coefficient found in the last experiment. The data, provided by the Italian Coast Guard, were colletcted during a SAR training: on November 13$^{th}$ 2013 at UTC 8:30$pm$ a dummy was seeded in the Sicily Channel and it was retrieved 13 hours later.

The geometrically scaled target and the statistic of the new forcings are included in the model; we estimate on the forcings a standard deviation 0.09 $m \cdot s^{-1}$ in current east component and 0.11 $m \cdot s^{-1}$ in the north one, while it is 4.07 $m \cdot s^{-1}$ in wind east component and 4.25 $m \cdot s^{-1}$ in the north one.

The data about latter experiment are shown in the Tab. 3 and 4.

**3 Results**

In this section the main results of each set of experiments are shown. The first set of experiments is about the configuration of the target and the effects of the corresponding jibing. Here the $unknown$ status, which has the highest error variance in the statistical calculation of the $leeway$, gives the best results; changing the jibing values the results don't change significatively. This is why we opt for the LEEWAY standard value 4%. The final map is shown in Fig. 2 where at the retrieval time the target is outside the particles cloud. The distance between the mass center of each seeding group and the retrieval point/time is shown in Fig. 3: we verify that the $3^{rd}$ group, seeded between the 9:30$am$ and 10:00$am$, is the closest to the target, with a distance which oscillates around the value of 9 km; the smallest absolute distance (about 2 $km$) is reached 31.5 hours later relative to the retrieval time by the $4^{rd}$ group one, seeded between the 10:00$am$ and 10:30$am$.

The second set of experiments studies the influence of the perturbation coefficients on the results. These coefficients have been added in the regression line used to calculate the $leeway$, to include the variance increasing with the wind speed. In our experiments we verify that the spread is such that the particles final cloud includes the target at retrieval time (Fig. 4) when the slope is $a_n$=a+$\epsilon_n$/5 and the offset is $b_n$=b+2 ·$\epsilon_n$, i.e. the perturbation coefficients are 4 times greater than the ones proposed in Breivik and Allen (2008) . The $3^{rd}$ group is again the nearest to the retrieval point and the relative distance oscillates around the value of 5.5 $km$ (Fig. 5); such group has the absolute smallest distance (about 4.3 km) 5 hours later relative to the retrieval time. As a result the uncertainty about the time of the accident is reduced and the distance between the mass center of the favoured seeded group and the retrieval point is almost halved but the dispersion is now very higher.

To reduce the dispersion and to give a greater weight to the advection, a different option about the perturbation coefficients is checked: if wind velocity is lower than 10 $m·s^{-1}$ we increase only the offset perturbation coefficient whereas for wind velocity greater that 10 $m · s^{-1}$ we increase only the slope one, according to different weight of the offset/slope of the regression line at lower/higher wind speeds. Now the particles cloud includes the target at the rescue time and the dispersion of the particles is reduced (Fig. 6) but the distance between the mass center of the $3^{rd}$ group and the retrieval point/time is very similar to that one in the first experiment, resulting be about 8.9 $km$ (Fig. 7). As in the first experiment, the $4^{rd}$ seeded group has again the smallest absolute distance (about 2.8 $km$) 31.5 hours later relative to the retrieval time.

The previous experiments pointed out the different performances of the model when the real statistic on forcings and the statistical regression line to calculate the $leeway$ velocity are used. Not only the heteroscedasticity of the experimental dataset compiled by Allen and Plourde (1999) needs to be correctly accounted for to optimize the model performances, but we also need to choose the best perturbation coefficients to use in the regression line. We don't know them in advance and we only can say that they should be such that the dispersion doesn't swamp the advection role and at the same time it should induce to maximize the POC.

The third set of experiments consists in simulating the PIW drift by means of the $almost$ deterministic law. After checking the projected frontal area by means of the allometric parameters for a "standard human body" according to Herman (2006), we choose a height/width ratio equal to 4.44. The final critic Reynolds number range is estimated between 165000 and 168000. The Reynolds numbers height based are always between O($10^4$) and O($10^5$); due to these values and to roughness role on

the boundary layer modifications (Yeo and Jones, 2011) we conclude that the PIW boundary layer in our experiment is often laminar with transition to turbulence prevalently precritic. That means that the drag crisis never occurs, the boundary layer never separates and the drag coefficient in the turbulent boundary layer is constant for a large Reynolds numbers range. In our experiments the drag coefficient corresponding to the best results is 1.12. All the parameters to solve the forces balance
5  equation are in the Tab. 2.

The results now show that at retrieval time the cloud includes the target (Fig. 8) and at same time the dispersion of the particles looks low and qualitatively comparable with that one in Fig. 6 where the $leeway$ perturbation coefficients have been setted separately on the slope or on the offset of the regression line according to wind velocity. The absolute distance of $3^{rd}$ group of particles from the rescue point is minimum (Fig. 9) and is about 1.3 $km$.

10  In the figure 10, where the snapshots of figure 9 are showed without the probability map, it is visible that the particles move according to surface hydrodynamic; some mesoscale and sub-mesoscale cyclonic and anticyclonic structures are visible, according to complex circulation of the area as described in Pascual et al. (2013), Rinaldi et al. (2010), Iacono et al. (2013), Astraldi and Gasparini (1994). In particular, the area where the particles are seeded is characterized by three principal surface structures: two cyclonic gyres (named "A" and "C"), separated by an anticyclonic gyre (named "B"). The cyclones "A",located
15  in the area $9.85° − 11.3°$E, $39.3° − 40.25°$N, and "C", located in the area $10° − 10.75°$E, $40.5° − 41.1°$N are persistent during the period of the simulation and show circulation features according to summer time ones as described in Rinaldi et al. (2010) and Marullo et al. (1994). The most of particles are seeded along the western brunch of the gyre "A" while the smallest amount is seeded between the cyclonic gyre "C" and the antyciclonic gyre "B". On the time the particles set is separated in two subsets: the gyre "A" track the larger subset westwards near the Sardinia coast while the variability of the current between the gyres
20  "C" and "B" tracks the remaining particles to the open sea. Due to this analysis we conclude that the persistent summer gyre "A" is responsable for the motion of PIW in our experiment and we think that in all probability the person fell in water north of the retrieved point.

We checked our approach in a different area (Sicily Channel), using experimental data (Tab. 3) and scaling geometrically our target (Tab. 4). The results show that the target is inside the particles cloud, which minimum distance, estimated about in 5.8
25  $km$, is reached at the rescue time (Fig. 11). The target is located on the external border of the cloud (Fig. 12), then the POC is from medium to low value. The figure 13 shows that the particles cloud is seeded on the external part of an anticyclonic vortex, in an area where the current flowing along the southern Calabria coastline interacts with the current coming from the Messina Strait, separating the Sicily Island from the Italian Peninsula. It is a narrow passage where the Tyrrhenian and Ionian Sea sub-basins are interconnected: the interaction between its bathymetry and topography and the strong currents induces the
30  generation of inertial eddies and strong horizontal current shears, generally located at along both Sicily and Calabria main capes (Cucco et al., 2016). Although the Mediterranean Sea is characeterized by very small tidal displacements which don't influence significatively the circulation (Sannino et al., 2015), in the Messina Strait large gradients of tidal displacement are registered because the semidiurnal tides in the Tyrrhenian and Ionian Sea are approximately in phase opposition. The tides then in the Messina Strait are the principal forcing of the circulation which develops mainly along the main axis of the channel; in
35  particular, during the flood the flow is northward whereas during the ebb the flow is southward (Cucco et al., 2016).

Our experiment was performed during the ebb phase; the dummy was seeded in an area where northern border of the anticyclonic vortex, located in the area $15.3° − 16.3°$E, $37.2° − 37.65°$N, mixes with both the surface flow carrying the Ionian waters from the southern Calabria coastline to Sicily eastern coastline and part of the current coming from Messina Strait. The area is therefore characterized by an high turbulence which plays an important role on the particles dispersion.

In general, the figure 10 such as the figure 13 underline the importance to have an efficient operational ocean prediction system for the SAR activities not only because it is the forcing of the dispersion lagrangian model, but also because it gives many informations useful to analyze the particles cloud final evolution. In particular, in the figure 13, it seems that approximation on the POC calculation is greater than that one estimated in the Sardinia experiment but we think that it can be improved if the set of particles is clustered according not only to seeding time but also splitting the cloud in subsets as much as are the hydrodynamic structures; in this way each subset will have an own POC which could be high even if the it includes a small number of particles. We are currently working to this hypothesis.

**4   Conclusions**

We have presented the results of numerical experiments performed to estimate the Probability of Containment (POC) of Person in Water (PIW). We have referred to a real event occurred in the western Thyrrenian sea. The LEEWAY model, nested with the sub-regional hydrodynamic model TSCRM has been used. The real statistic of the forcings on the observation period was accounted for and the Last Known Position has been based on the trajectory of the ferry during 3.5 hours, recorded by the Automatic Information System; to reproduce the event a set of 1000 particles seeded in 8 time step has been used.

The objective of the work has been to validate an *almost* deterministic law to calculate the *leeway*, based on the boundary layer theory; these results have been compared with those obtained estimating the *leeway* by means of the standard statistical approach. The latter has produced the best results when the perturbation coefficients in the regression line have been changed separately on the offset or on the slope of the regression line according to wind velocity.

To calculate the *leeway* in the *almost* deterministic approach the PIW has been cylinder-shaped with a height/width ratio equal to 4.44; the critic Reynolds number has been found in the range between 165000 and 168000 and the drag coefficients have been estimated equal to 1.12. Now the results have showed that at the retrieved time the particles cloud includes the target and the $3^{rd}$ subset of particles is representative of the event then we also can estimate the time of the accident with good approsimation. The results estimated by means of the standard statistical approach are similar but with the new approach the distance of the mass center of the favoured subset of particles is strongly reduced 6.5 times, resulting of $1.3km$. The tests in Sicily Channel using a similar target have confirmed the reliability of the method.

A second important result is that the particles distribution in both experiments is coherent with the hydrodynamic structures, highlighting the importance to have an efficient operational ocean prediction system for the SAR activity: the hydrodynamic field is required not only because it forces the lagrangian model but also because it allows to read the final results in a more full way. We think infact that the results about the POC can be improved splitting the set of particles in subsets as much as are hydrodynamic structures, so that each subset can significatively contribute with the own POC; it corresponds to consider

time-evolving probability density functions of the location of the search object as much as are the hydrodynamic structures. At last, the Stokes drift will have also to be included.

The results here obtained encourage to explore other different target categories. We believe that on the long time it will allow to overcame the limitations connected with the standard approach based on the tabled statistical parameters, providing leeway-drift formulas for practical use based on appropriate Reynolds critic numbers and drag coefficients for specific targets. This idea is supported by the realistic option to execute tests in naval tank, simulating different meteomarine conditions, scaling opportunely specific targets and then overcoming the costs and difficulties to acquire field data.

*Acknowledgements.* We would like to thank to C.te Sirio Faé of Italian Coast Guard for providing the data about our experiments and for supporting in the interpretation of the IAMSAR Manual. Thanks also to collegues of the CNR-INSEAN: Massimo Miozzi for his constructive comments during the work and Paola Carratú for her special contribution.

This study was supported by the Italian Project PON-TESSA (PON01-02823) and by RITMARE (Ricerca ITaliana per il MARE) Flagship Project of Italy (PNR 2011-2013,approved by CIPE with adjudication 2/2011 on 23/03/2011), funded by the Ministry for Education, University and Research of Italy - MIUR.

[revised manuscript text omitted]

---

## Referee Comment (RC2) · Anonymous Referee #2 · 9 Jul 2016

General Comments: The manuscript "Evaluation of the Search and Rescue leeway model into the Tyrrhenian sea: a new point of view" by A. Di Maio et al. deals an important issue of human and socio-economic relevance. The optimisation of the SAR operations is a day-to-day increasing need.

The Authors propose an innovative leeway calculation, which is tested against the standard and widely used approach of Breivik and Allen (2008) through a comprehensive set of numerical experiments and a real PIW case and a dummy experiment field data .

The manuscript is well structured and nicely written. It is concise and the results are robust. The English has been much improved after including the corrections suggested by Reviewer #1. Those suggestions have also improved other aspects of the

manuscript.

Therefore I recommend the manuscript for publication, but I would like the Authors to consider some suggestions and to correct some typos. Further I refer to the updeted version, uploaded as SP on

Minor remarks:

1. I think that a Figure including the domains of the ocean models used would be illustrative for the reader.

2. Pag 8, line 13: The perturbation coefficients are 4 times LARGER than the ones proposed by Breivik and Allen (2008). I think this result merits a more in depth discussion, as it shows the unavoidable shortcomings of a very general statistical approach.

3. Authors very correctly note the importance of tidal dynamics in the Messina Strait and surrounding areas, however it is not clear to me if the TSCRM includes tidal forcing. This should be clarified.

4. In Figures 3,5,7,9 it is difficult to distinguish the lines, it appears as there is a single lines colour and style for every two sets.

Typos: Title: into the Tyrrhenian...> in the Tyrrhenian..

P1, L3: stocastic > stochastic P1, L17: Safety and assets > Safety of lives and assets P.1 L18: sea > sea state P1, L20: life > lives P3, L16: main > aim P3, L17: targest > targets P3, L26: (PIW). > (PIW) case. P4, L5: used tp produce > used to produce P4, L7: transmitting > providing P4, L19: foundamental > fundamental P5 : I do net see the need of introducing two new acronyms (DWL, CWL) when you can use Ld and Lc P5, L19: Know > Known P6, L10: geographycal > geographical P6, L10: arbitrarly > arbitrarily P7,L1. Know > Known P7, L3: Tabled configurations (if you refer to a previous work, cite it here please). P7, L25: geographycal > geographical P7, L29: statistic > statistics P8, L4: ...the results don't change significatively > ...does not change significantly the results P8, L7: am > pm in both cases P8, L9: am > pm in both cases

P8, L17: very higher > much higher P8, L20: weight > weights P8, L25: statistic > statistics P8, L28: don't > do not P8, L29: doesn't > does not P9, L11: hydrodynamic > circulation patterns P9, L17: The most > Most P9, L17: brunch > branch P9, L19: track > drives P9, L20: track > drives P9, L20: Due to this analysis we conclude > This analysis allows us to conclude P9, L30: at along > along P10, L3: coastline > coast (both) P10, L5: such as the > as well as P10, L6: because it is the > as a P10, L7: informations > information P10, L11: working to this hypothesis > working to test this hypothesis

---

## Author Comment (AC2) · 27 Jul 2016

Thank you to referee#2 for his constructive comments about the manuscript, especially with reference to the comment number 2 with which we are absolutely agree. We have deepened the study about the statistical approach inlcuding a new test: that allow us to find a regression lines beam corresponding to a good simulation of the real event . Setting correctly the standard error coefficents, the distance between the particles and the rescue point/time is reduced and the dispersion is low. The pattern is now comparable to the results obtained by means of the new approach; anyway the latter provides the absolute smallest distance and that seems emphasize even more the goodness of the method.

The pdf file attached here is the new version of the manuscript containing more details

in the results discussion. New figures are included

A 2nd pdf file with reply to each comment is also attached

Please also note the supplement to this comment:
http://www.nat-hazards-earth-syst-sci-discuss.net/nhess-2016-109/nhess-2016-109-AC2-supplement.zip

———————————————————